# Expression and Impact of Vaspin on In Vitro Oocyte Maturation through MAP3/1 and PRKAA1 Signalling Pathways

**DOI:** 10.3390/ijms21249342

**Published:** 2020-12-08

**Authors:** Patrycja Kurowska, Ewa Mlyczyńska, Anthony Estienne, Alix Barbe, Iwona Rajska, Katarzyna Soból, Katarzyna Poniedziałek-Kempny, Joelle Dupont, Agnieszka Rak

**Affiliations:** 1Department of Physiology and Toxicology of Reproduction, Institute of Zoology and Biomedical Research, Jagiellonian University in Krakow, 30-387 Krakow, Poland; patrycja.kurowska@doctoral.uj.edu.pl (P.K.); ewa.mlyczynska@doctoral.uj.edu.pl (E.M.); 2INRAE, UMR85, Unité Physiologie de la Reproduction et des Comportements, 37380 Nouzilly, France; anthony.estienne@inra.fr (A.E.); alix.barbe@inra.fr (A.B.); joelle.dupont@inra.fr (J.D.); 3Department of Reproductive Biotechnology and Cryopreservation, National Research Institute of Animal Production, 32-083 Balice, Poland; iwona.rajska@izoo.krakow.pl (I.R.); katarzyna.sobol@izoo.krakow.pl (K.S.); katarzyna.kempny@izoo.krakow.pl (K.P.-K.)

**Keywords:** oocyte maturation, vaspin, ovary, adipokines, MAP3/1, PRKAA1, pig

## Abstract

Oocyte maturation is a critical stage in embryo production and female reproduction. The aims of this study were to determine: (i) the mRNA and protein expression of vaspin and its receptor 78-kDa glucose-regulated (GRP78) in porcine cumulus–oocyte complexes (COCs) by real-time PCR and Western blot analysis, respectively, and their localisation by immunofluorescence; and (ii) the effects of vaspin on in vitro oocyte maturation (IVM) and the involvement of mitogen ERK1/2 (MAP3/1)- and AMPKα (PRKAA1)-activated kinases in the studied processes. Porcine COCs were matured in vitro for 22 h or 44 h with vaspin at a dose of 1 ng/mL and nuclear maturation assessed by Hoechst 33342 or DAPI staining and the measurement of progesterone (P4) level in the maturation medium. We showed that vaspin and GRP78 protein expression increased in oocytes and cumulus cells after IVM. Moreover, vaspin enhanced significantly porcine oocyte IVM and P4 concentration, as well as MAP3/1 phosphorylation, while decreasing PRKAA1. Using pharmacological inhibitors of MAP3/1 (PD98059) and PRKAA1 (Compound C), we observed that the effect of vaspin was reversed to the control level by all studied parameters. In conclusion, vaspin, by improving in vitro oocyte maturation via MAP3/1 and PRKAA1 kinase pathways, can be a new factor to improve in vitro fertilisation protocols.

## 1. Introduction

Oocyte maturation is a critical stage of embryo production in mammals, characterised by periods of meiotic arrest and resumption [1]. Balance is needed between factors promoting or inhibiting oocyte maturation for subsequent fertilisation and embryo development [1]. In vivo, oocytes remain at the immature germinal vesicle (GV, prophase of meiosis) stage to luteinising hormone (LH) ovulatory peak [2], but oocytes may also resume meiosis outside the body. In vitro maturation includes both nuclear and cytoplasmic maturation. Nuclear maturation is the ability of oocytes at the GV stage to complete metaphase-I, transit to metaphase-II and in the production of first polar body connected to their higher progesterone (P4) production [3]. Cytoplasmic maturation is characterised by molecular and structural changes, including reorganisation of the cytoskeleton to provide a mature ovum to support fertilisation and early embryonic development [4]. Oocyte maturation is regulated by two opposite kinases: mitogen-activated kinase (MAP3/1) phosphorylates cytoskeletal proteins and has a key role in meiotic cell division, while AMP-activated kinase (PRKAA1) has an opposite effect—maintenance of the meiotic block in porcine and bovine oocytes [5,6]. It is important that the in vitro environment must support both nuclear and cytoplasmic maturation; knowledge about factors stimulating oocyte maturation in vitro is a first step to improve in vitro fertilisation protocols. As described previously, adipokines such as leptin and adiponectin stimulate porcine oocyte maturation in vitro [7,8,9]. More precisely, Chappaz et al. [9] showed the expression of adiponectin receptors in porcine oocytes and cumulus cells, as well as a decrease in immature oocyte number by adiponectin addition via the p38MAP pathway. Interestingly, Craig et al. [7] noted upregulation in leptin receptor protein expression after oocyte maturation and the stimulatory effect of leptin on this process via the MAPK pathway.

It is known that the reproductive success of animals, including pigs, depends on nutritional status and energy resources. The relevant energy status of sows determines the normal reproductive functions; e.g., sows with low weight require a longer time to first oestrus and their offspring are less numerous and characterised by low birth weight [10]. Moreover, adipose tissue produces multiple hormones called adipokines, which play a pleiotropic function in the body, including ovarian function regulation [11]. Currently available data suggest that adipokines (leptin, adiponectin, apelin and chemerin) may operate as paracrine mediators linking oocyte maturation, early embryo development, and implantation in different species including pigs, bovines, and goats [7,9,12,13,14,15]. Recent data suggest that another adipokine—vaspin (visceral adipose tissue-derived serine protease inhibitor) is known as a regulator of energy balance, to decrease food intake [16], promote preadipocytes differentiation [17], improve insulin sensitivity and glucose tolerance [18], and play an important role in reproduction. In our previous published data, we described the dependence on oestrous phase vaspin expression of porcine ovarian structures, ovarian follicles [19], and corpus luteum (CL) [20]. Interestingly, we showed that the plasma vaspin level is dependent on fattening: a higher expression was noted in fat Meishan pigs compared to lean Large Whites [19]. Similarly, Barbe et al. [21] showed that in both plasma and perirenal white adipose tissue, the amount of vaspin was higher in Meishan pigs. Moreover, the expression of vaspin is strongly dependent on hormones, which are connected to ovarian follicle growth and development, ovulation, CL formation and regression, and levels of gonadotropins and steroid hormones such as oestradiol, testosterone and P4, insulin and prostaglandins E and F2α [19,20]. Furthermore, vaspin regulates ovarian physiology by a direct stimulatory effect on phosphorylation of multiple kinases MAP3/1, protein kinase A (PKA), Janus kinase (STAT3) and PRKAA1 [20,22], steroidogenesis [22] proliferation, and inhibition of apoptosis in ovarian follicle cells [23]. In porcine CL cells, vaspin enhances steroidogenesis by increasing level of P4 and oestradiol, as well as enzymes participating in its synthesis via 78-kDa glucose-regulated protein (GRP78) receptors and the PKA signalling pathway. Moreover, vaspin exerts a positive impact on CL angiogenesis connected with upregulation in mRNA expression and the secretion of vascular and endothelial growth factors and angiopoietin 1 by GRP78 and MAP3/1 activation [24]. Importantly, the effect of vaspin on oocyte maturation has not been studied in pigs or other species. Based on previous findings concerning the role of vaspin in the porcine ovary and the connection of adipokines to oocyte in vitro maturation, we hypothesise that vaspin and GRP78 are expressed in porcine oocytes and vaspin has a stimulatory effect on in vitro oocyte maturation.

Thus, we studied: (i) vaspin, GRP78 mRNA, and protein expression in oocytes and cumulus cells before (0 h) and after in vitro maturation (44 h), as well as vaspin and GRP78 immunofluorescence in cumulus–oocyte complexes (COCs), (ii) the effect of vaspin on in vitro oocyte maturation by measuring the percentages of oocytes in metaphase-II of meiosis and measuring P4 levels in the oocyte maturation medium, (iii) the effect of vaspin on MAP3/1 and PRKAA1 kinase phosphorylation in oocytes and cumulus cells, and (iv) the involvement of MAP3/1 and PRKAA1 kinases in the action of vaspin on in vitro oocyte maturation.

## 2. Results

### 2.1. Vaspin mRNA and Protein Expression before and after In Vitro Oocyte Maturation, as well as Its Immunolocalisation in COCs

We investigated vaspin mRNA and protein expression before (0 h) and after (44 h) oocyte in vitro maturation, as well as its immunolocalisation in COCs. We showed that vaspin mRNA transcription decreased significantly after 44 h of in vitro maturation in oocytes and cumulus cells, by 6.2- and 3.6-fold, respectively (*** *p* < 0.001, Figure 1A). On the other hand, the protein levels of vaspin in oocytes and cumulus cells were significantly higher after oocyte maturation (*** *p* < 0.001, Figure 1B). Additionally, strong fluorescence signals for vaspin in COCs and denuded oocytes were detected. No immunoreaction was observed for the negative controls (Figure 1C).

### 2.2. GRP78 mRNA and Protein Expression before and after Oocyte In Vitro Maturation, as well as Its Immunolocalisation in COCs

Hence, we examined GRP78 mRNA and protein expression before (0 h) and after (44 h) oocyte in vitro maturation, as well as its immunolocalisation in COCs. We observed that GRP78 mRNA expression was upregulated after in vitro maturation in both oocytes and cumulus cells by 3- and 4-fold, respectively (*** *p* < 0.001, Figure 2A). A similar effect was observed for GRP78 protein level (** *p* < 0.01, *** *p* < 0.001, Figure 2B). Moreover, as shown in Figure 2C, we observed strong fluorescence signals for GRP78 in COCs and denuded oocytes. No immunoreaction was observed for the negative controls (Figure 2C).

### 2.3. Effect of Vaspin on Nuclear In Vitro Oocyte Maturation and P4 Release in the In Vitro Oocyte Maturation Medium

We investigated the effect of vaspin at a dose 1 ng/mL on in vitro porcine oocyte maturation. Oocyte maturation was identified by nuclear maturation status by Hoestch 33342 or DAPI staining (Figure 3A), and we analysed P4 level in culture medium by ELISA. For the control group, 40.5% of oocytes underwent metaphase-I stage after 22 h of maturation, while for vaspin, it was 66.3% (* *p* < 0.05, Figure 3B). Stimulatory effect was confirmed by the P4 level in culture medium (26.540 ± 0.57 ng/mL) compared to control (23.51 ± 0.43 ng/mL; ** *p* < 0.01, Figure 3C). Vaspin also upregulated in vitro oocyte maturation after 44 h of culture: about 75.6% of oocytes had progressed to the metaphase-II stage compared to 58.7% in the control (* *p* < 0.05, Figure 3D). We observed that vaspin significantly increased P4 secretion by COCs (36.07 ± 0.44 ng/mL) compared to the control (32.62 ± 0.63 ng/mL) after 44 h of culture (** *p* < 0.01, Figure 3E).

### 2.4. Effect of Vaspin on MAP3/1 and PRKAA1 Kinase Phosphorylation in Oocytes and Cumulus Cells after In Vitro Maturation

We determined the effect of vaspin at 1 ng/mL on MAP3/1 and PRKKA1 kinase phosphorylation in oocytes and cumulus cells after 44 h of in vitro culture. As shown in Figure 4, addition of vaspin to the culture medium for 44 h significantly increased MAP3/1 (** *p* < 0.01 and *** *p* < 0.001, Figure 4A), whereas it decreased PRKAA1 (** *p* < 0.01 and *** *p* < 0.001, Figure 4B) phosphorylation in both oocytes and cumulus cells, confirming the stimulatory effect of vaspin on oocyte maturation.

### 2.5. Involvement of the MAP3/1 and PRKAA1 Kinases in the Effect of Vaspin on In Vitro Oocyte Maturation

Based on the result that vaspin modulated the phosphorylation of kinases MAP3/1 and PRKAA1, in the next set of experiments, we examined the role of these kinases in the vaspin-mediated effect on oocyte maturation. Oocyte nuclear maturation and P4 concentration in culture medium were assessed after 44 h incubation with pharmacological inhibitors of MAP3/1 (PD98059 at 100 μM) or PRKAA1 (Compound C at 1 μM), which indicated that simultaneous treatment with PD98059 or Compound C with vaspin (1 ng/mL) reversed nuclear maturation and P4 levels compared to control (Figure 5, statistical analysis carried out at *p <* 0.05).

## 3. Discussion

Oocyte maturation is a key event in a mammal’s embryo production and is essential for subsequent fertilisation [1]. Interestingly, increasing evidence has indicated vaspin as an important factor in female fertility, linking ovarian physiology with energy resources [19,20]. In the present study, we expand the knowledge about the role of vaspin in reproduction by studying the expression and impact of vaspin on porcine oocytes. We showed that (i) vaspin and GRP78 mRNA and protein expression changed after COC maturation in oocytes and cumulus cells, (ii) vaspin enhanced significantly porcine oocyte maturation and P4 secretion by COCs, (iii) vaspin upregulated MAP3/1 and decreased PRKAA1 phosphorylation, and iv) the stimulatory effect of vaspin on oocyte maturation was mediated by MAP3/1 and PRKAA1 kinase activation.

Knowledge of specific genes or proteins and their expression patterns in the early stages of porcine embryogenesis is still limited. Investigation of pig oocyte transcriptome or proteome and changes during in vitro maturation has shed more light on the mechanisms of this process. Previous studies have shown that vaspin expression depended on oestrous cycle stage in the porcine ovary [19,20], but its presence in germ cells has not been studied. For the present study we chose our research model based on previous papers describing effects on different hormones on porcine in vitro oocyte maturation [9,25]. Moreover, Marchal et al. [26] identified that even if prepubertal gilt oocytes appeared less competent and meiotically than adult sows, they can be used to produce blastocysts able to develop to term. Our present data provide novel evidence about vaspin and GRP78 gene transcripts and protein level in the oocytes and cumulus cells before and after in vitro maturation. Here, we showed that vaspin mRNA amounts decreased, while its proteome, as well as GRP78 mRNA and protein, was upregulated after in vitro maturation. The same pattern of vaspin and GRP78 was observed for oocytes and cumulus cells, so we could hypothesise that there are no immunolocalisation changes during in vitro maturation. However, this remains to be determined. The present findings indicate that its expression is dependent on oocyte maturation and suggests potential autocrine and/or paracrine effects of vaspin in this process. Differences between vaspin mRNA and protein levels are probably linked to translational regulations. Poor genome-wide correlation between expression levels of mRNA and protein are commonly reported [27] and are explained by complicated post-transcriptional mechanisms involved in transcribing mRNA into protein. Our results obtained are in a good agreement with previous data of Ellederova, who showed analogous changes in GRP78 receptor expression in in vitro cultured porcine oocytes as analysed by two-dimensional gel electrophoresis and mass spectrometry [28]. Moreover, stimulation in GRP78 expression can be linked to vaspin protein upregulation. Similar interactions between a protein and its receptor have been observed in bovine embryos: briefly, a high dose of leptin (10 ng/mL) increased leptin receptor expression [29]. Furthermore, changes in adipokine expression before and after in vitro maturation are well documented in the literature: e.g., adiponectin receptor 2 transcription decreased during bovine in vitro maturation in COCs, with no differences in adiponectin level [30]; a similar downregulation was observed for visfatin in the oocytes [31]. Interestingly, there is a link between adipokine system expression and its effect on in vitro oocyte maturation; for instance, adiponectin decreases epidermal growth factor (EGF) and foetal calf serum (FCS)-induced oocyte maturation [30]. Based on the above, we hypothesise that increases in vaspin and GRP78 protein levels are connected with a stimulatory effect on oocyte maturation.

Oocytes at GV removed from the ovarian follicle and cultured in vitro may spontaneously resume meiosis, understood as completing the metaphase-I transit to metaphase-II (where it is arrested until fertilisation) and producing the polar body [3]. As we anticipated, we showed the stimulatory effect of vaspin (1 ng/mL) on nuclear oocyte maturation. The total time to undergo metaphase-II for porcine oocytes cultured in vitro is 44 h [32]. In our experiments, we checked the progress in oocyte maturation by counting the percentage of oocytes first in metaphase-I after 22 h of culture (percentage of oocytes in metaphase-I) and then at metaphase-II after 44 h. The positive effect of vaspin on oocyte maturation was confirmed by increases in P4 levels in the culture medium after 22 and 44 h of culture. During in vitro maturation, P4 is produced by cumulus cells; moreover, in bovine oocyte culture, P4 addition stimulates meiotic reinitiation [33]. Thus, vaspin could have an effect on porcine oocyte maturation through an upregulation of P4 production by COCs. This is consistent with our previous findings, where we showed that vaspin (0.01–10 ng/mL) enhanced P4 secretion by ovarian follicles cells cultured in vitro [22]. The ovarian follicle creates an environment for the growth and development of oocytes, so vaspin can stimulate oocyte maturation, not only by a direct effect on P4 production by COCs, but also by an increased P4 level in porcine ovarian follicles. Furthermore, we showed previously the positive effect of vaspin on P4 production by the porcine CL, which is another confirmation of its stimulatory role in P4 synthesis [20]. On the other hand, it has been suggested that only high doses of P4 (500 ng/mL) stimulate in vitro porcine oocyte maturation, with no effect at a lower dose (50 ng/mL) [34]. In our experiments, the increase in P4 level was not comparable to these data and consequently could result from an increase in the percentage of mature oocytes, suggesting a potential direct action of vaspin on oocyte nuclear maturation. Furthermore, some authors observed that, in porcine oocytes, isoflavone daidzein inhibited P4 production without affecting oocyte in vitro maturation [35]. It is undeniable that vaspin stimulates in vitro oocyte maturation, but whether this is a direct or an indirect effect through increasing P4 production requires future research. Interestingly, the important and various effects of adipokines on oocyte maturation has been described previously: chemerin induced an arrest at the GV stage of bovine oocytes and decreased P4 level [13], while adiponectin accelerated the meiotic maturation of porcine oocytes [9].

Additionally, in the present study we showed that vaspin increased the phosphorylation of MAP3/1 kinase in oocytes and cumulus cells and had an opposite effect on PRKAA1. Previously, we have shown the rapid, time-dependent stimulatory effect of vaspin at a dose of 1 ng/mL on the phosphorylation of both kinases in ovarian follicle cells [22]. The differences observed for PRKKA1 can be dependent on culture time or the type of structure studied. Literature data show that MAP3/1 and PRKAA1 play important roles in porcine oocyte maturation, which may confirmed by the direct involvement of vaspin in that process. MAP3/1 phosphorylates cytoskeletal proteins and nuclear lamin and plays a key role in meiotic cell division [5]. During meiosis, MAP3/1 is activated around the GV breakdown stage with a peak during metaphase-II and participates in cumulus cell expansion [36]. On the other hand, the level of PRKAA1 phosphorylation is high in immature oocytes and ovarian cumulus cells, while reduced during maturation in porcine COCs [37]. Is it a well-known fact that MAP3/1 phosphorylation increases during oocyte maturation after 24 h in bovine [13] and after 44 h in porcine [7] COCs, while PRKAA1 phosphorylation decreases in bovines after 24 h [6], so in our research we focused just on the direct effect of vaspin, without studying time-dependent relationships. Interestingly, some factors are able to modulate oocyte maturation by altering the phosphorylation of the mentioned kinases: for example, the PRKKA1 activation in response to metformin-blocked meiotic progression at the GV stage in bovine [6] and porcine [37] oocytes. Finally, the connection of adipokines to MAP3/1 and PRKAA1 phosphorylation in COCs has been described. The inhibitory effect of chemerin [13] and apelin [14] on bovine oocyte maturation has been linked to its downregulatory action on MAP3/1 phosphorylation. Based on that, the effect of vaspin on the phosphorylation of key kinases in oocyte maturation is another confirmation of its positive, most likely direct, action on this most important event in reproduction.

Finally, we examined the molecular mechanism of the action of vaspin on in vitro maturation of porcine oocytes. Using pharmacological inhibitors of MAP3/1 (PD98059) and PRKAA1 (Compound C), we indicated that the stimulatory effect of vaspin on oocyte nuclear maturation measured by its abundance in metaphase-II of meiosis, as well as P4 concentration after 44 h of culture, was reversed to the control level. We did not observe significant differences in oocyte maturation after addition of inhibitors, compared to the control. Our findings are in good agreement with those of Bilodeau-Goeseels, who did not observe stimulation of porcine oocyte maturation after Compound C addition [38]. Similarly, maturation of porcine oocytes cultured with leptin alone or with MAP3/1 inhibitor was not decreased under control level [7]. However, Sirotkin et al. [8] showed that PD98059 inhibitor alone dramatically decreased the percentage of oocytes undergoing maturation. Macroscopically, we observed inhibition of cumulus cell expansion by MAP3/1 inhibitor. Nevertheless, the stimulatory action of vaspin on the percentage of oocytes in metaphase-II was inhibited, which clearly identified the molecular mechanism of its action. Furthermore, the present data are in a good agreement with findings indicating that pharmacological inhibition of MAP3/1 inhibited the stimulatory action of leptin [7] and adiponectin [9] on porcine oocyte maturation.

Our results are important for female fertility because if in vitro maturation is not carried out in a precise manner under optimal conditions, subsequent fertilisation and embryo development will be compromised [1]. Interestingly, vaspin’s link with ovulation has been shown previously: Dogan et al. [39] compared serum vaspin levels in polycystic ovarian syndrome in women with either failed or successful ovulation induction and measured significantly lower vaspin levels in responders achieving ovulation, indicating this adipokine to be a predictor. Vaspin levels in porcine ovarian follicles and serum are about the same (1 ng/mL) [19]. Moreover, in our previous research, vaspin expression in adipose tissue, ovary, serum and follicular fluid were found to show higher expression in each studied sample from Meishan pigs characterised by higher fattening levels [19]. In a previous study we indicated that vaspin stimulated porcine ovarian steroidogenesis [22] and proliferation and inhibited apoptosis [23] in a dose-dependent manner. Interestingly, several studies have shown that the elevated vaspin levels observed in obesity play a compensatory role in the organism by improving glucose tolerance [40], by its cardioprotective function [41] and by decreasing food intake [16]. We documented that in the ovary, vaspin also acts as a positive reproductive regulator, including its stimulatory action in in vitro porcine oocyte maturation, which could probably compensate for infertility events caused by obesity.

## 4. Materials and Methods

### 4.1. Reagents

Electrophoresis marker and TRIzol reagent were purchased from ThermoFisher Scientific (Waltham, MA, USA). Medium TCM199 (product no. M4530), kanamycin sulfate from streptomyces kanamyceticus (product no. K1377), mineral oil (M5310), L-cysteine (product no. C7352), LH (product no. L6420), follicle stimulating hormone (FSH) (product no. F4021), fibroblast growth factor (FGF) (product no. F3133), EGF (product no. E1257), butryl AMP (dbcAMP) (product no. D0260), hyaluronidase (product no. H3506), vaspin (product no. SRP4915), Compound C (product no. P5499), Triton X-100 (product no. T8787), glycine (product no. G2879), Tris, phosphate-buffered saline (PBS) (product no. D4031), Tween 20, Laemmli buffer (product no. 38733), FCS (product no. F4135) and polyvinylpyrrolidone (PVP; product no. PVP360) were obtained from Sigma-Aldrich (St. Louis, MO, USA). PD98059 (product no. 1213) was obtained from Tocris (Bristol, GB). The 4–20% gels (product no. 456-1093) and membranes (product no. 1704156) were obtained from Bio-Rad (Hercules, CA, USA). Paraformaldehyde (product no. 11699408) was obtained from VWR International (Radnor, PA, USA).

### 4.2. COC Collection

Porcine ovaries were collected from prepubertal gilts (4–5 months old) [26] at a local abattoir as a byproduct under veterinarian control, ethics committee approval was not necessary. Ovaries were transported to the laboratory in PBS at 30–35 °C within 1 h of collection. Ovaries were washed three times in PBS with addition of 2% kanamycin at a temperature of 39 °C. Then, 3–6 mm-diameter follicles (15/ovary) were aspirated from sow (100/experiment) ovaries using a 12-gauge needle attached to a 10-mL disposable syringe [26]. Subsequently, oocytes were denuded by pipetting with 0.5% hyaluronidase and then collected using a glass pipette. To collect the cumulus cells, the solution was centrifuged (1000 rpm for 5 min). To determine vaspin and GRP78 basal mRNA expression, oocytes (*n* = 50/analysis) and cumulus cells (*n* = 50/analysis) were immediately frozen in liquid nitrogen and stored at −70 °C, while for protein level measurement, they were frozen at −20 °C. Oocyte and cumulus cell collections were repeated three times. Additionally, COCs (*n* = 50) were fixed in 4% paraformaldehyde for vaspin and GRP78 content analysis by immunofluorescence.

### 4.3. Porcine Oocytes In Vitro Maturation and Experimental Procedure

In vitro maturation was prepared according to the technique described by Poniedziałek-Kempny [32]. During one culture, COCs were obtained and accumulated from around 100 sows (15 aspirating follicles/ovary). Only COCs surrounded by a minimum of three cumulus cell layers (*n* = 100), with an evenly granulated cytoplasm were selected for maturation. COCs were divided in equal groups of 12–15 COCs based on macroscopic observation and cultured in 100 µL of maturation medium under mineral oil drops using petri dishes. Maturation medium consisted of TCM199 supplemented with 10% FCS [*v*/*v*], 1 mM L-cysteine [*v/w*], 5% porcine follicular fluid [*v*/*v*], 1 mM dbcAMP [*v/w*], EGF (10 ng/mL), FGF (40 ng/mL), FSH (10 IU/mL) [*v*/*v*], and LH (4.25 IU/mL) [*v*/*v*]. After 22 h of maturation, the oocytes were transferred to the same maturation medium but without EGF, FGF, FSH, LH, and dbcAMP for an additional 22 h under the same conditions. All cultures were maintained at 39 °C in a humidified atmosphere of 5% CO_2_/95% O_2_.

Experiment 1: To check vaspin and GRP78 expression after in vitro maturation, COCs were matured in maturation medium TCM199 for 44 h, and then cumulus cells were removed by gently pipetting of COCs in PBS supplemented with 0.5% hyaluronidase. Denuded oocytes (*n* = 50/analysis) and cumulus cells (*n* = 50/analysis) were frozen at −70 °C for vaspin and GRP78 mRNA and at −20 °C for protein expression analysis. The experiments were repeated three times.

Experiment 2: To determine the effect of vaspin on in vitro maturation of porcine oocytes or kinase phosphorylation, COCs were matured in maturation medium TCM199 in the presence or absence of recombinant vaspin at a concentration of 1 ng/mL, consistent with concentrations in porcine follicular fluid [19]. After 22 h and 44 h of culture, cumulus cells were removed from COCs by gently pipetting in of COCs PBS supplemented with 0.5% hyaluronidase. Denuded oocytes (*n* = 30/group) were fixed in 4% paraformaldehyde to assess nuclear maturation by DAPI or Hoechst 33342 staining. The medium was collected for P4 concentration measurement, while after 44 h of maturation a proportion of oocytes (*n* = 50/group) and cumulus cells (*n* = 50/group) were frozen at −20 °C to check pMAP3/1, MAP3/1, pPRKAA1, and PRKAA1 protein expression. Experiments were repeated three times.

Experiment 3: To investigate the molecular mechanism of action of vaspin on oocyte maturation, COCs were cultured in maturation medium TCM199 with MAP3/1 and PRKAA1 pharmacological inhibitors PD98059 at 100 μM and Compound C at 1 μM, respectively and vaspin at a dose 1 ng/mL. Doses of inhibitor were based on the literature [9,38]. After 44 h of maturation, cumulus cells were removed from COCs by gently pipetting of COCs in PBS supplemented with 0.5% hyaluronidase. Denuded oocytes (*n* = 30/group) were fixed in 4% paraformaldehyde for nuclear maturation analysis by DAPI staining, while the medium was stored for P4 secretion measurement. Experiments were repeated three times.

### 4.4. Real-Time PCR

Total RNA was isolated using TRIzol reagent according to the manufacturer’s protocol [13]. We performed reverse transcription as previously described [19]. Total RNA (1 μg) with 200 μM of deoxynucleotide triphosphate (Amersham, Piscataway, NJ, USA), 15 U of MMLV reverse transcriptase, 50 pmol of oligo(dT), 15.5 U of ribonuclease inhibitor, 75 mM KCl, 50 mM Tris-HCl (pH 8.3), 3 mM MgCl_2_, in a total reaction volume of 20 μL, was reverse-transcribed at 37 °C for 1 h. Then, cDNA dilution (1:5) was prepared. For the real-time PCR, 10 μL iQ SYBR Green Master Mix, 4.5 μL of water, 0.25 μL of each primer (10 μM) and 5.0 μL of template were used in a final volume of 20 μL. The cDNA was amplified using the MYIQ Cycler real-time PCR system (Bio-Rad) following the previously described protocol [42]. Glyceraldehyde 3-phosphate dehydrogenase (GAPDH) and actin were used as housekeeping genes and normalised according to rules described by Vandesompele [43]. The primer sequences were as follows: vaspin (forward 5′-GCTGTGAGTCGTGACCAAGT-3′ and reverse 5′-CACAGAGATGCTCCAAGGG-3′), GRP78 (forward 5′-ATCGAGTTGGCTTTCCGTGT-3′ and reverse 5′-CCAGTCAGTCAGTCAGCAGG-3′), GAPDH (forward 5′-GCACCGTCAAGGCTGAGAAC-3′ and reverse 5′-ATGGTGGTGAAGACGCCAGT-3′) and actin (forward 5′-ACGGAACCACAGTTTATCATC-3′ and reverse 5′-GTCCCAGTCTTCAACTATACC-3′).

### 4.5. Western Blot Analysis

Oocytes were quickly frozen and defrosted five times to destroy the zona pellucida, then oocytes and cumulus cells were lysed in Lamely buffe heated for 4 min at 95 °C. Electrophoresis, transfer and Western blot analysis were performed as previously described [44]. After blocking in 0.02 M Tris-buffered saline containing 5% bovine serum albumin (BSA) [*w*/*v*] and 0.1% Tween 20 [*v*/*v*], the membranes were incubated overnight at 4 °C with anti-vaspin, anti-GRP78, anti-pPRKAA1, anti-PRKAA1 antibodies (product nos. PA5-30989 (LOT# TK26665551N), PA5-19503 (LOT# GR32579381), PA5-17831 (LOT# TI2644134A) and PA5-17398 (LOT# TI2644137), respectively; Invitrogen, Carlsbad, CA, USA) or ani-pMAP3/1 and anti-MAP3/1 antibodies (product nos. 9101S (LOT# 28) and 9102 S (LOT# 27), respectively; Cell Signalling Technology, Massachusetts, DA, USA) 1:1000 at 5% BSA/TBST [*w*/*v*]. Next, the membranes were washed with TBST (Tris-buffered saline containing 0.1% Tween 20 [*v*/*v*]) and incubated for 1 h with a horseradish peroxidase-conjugated secondary antibody anti-rabbit (product no. #7074 (LOT# 28); Santa Cruz Biotechnology, Dallas, TX, USA) 1:1000 at 5% BSA/TBST [*w*/*v*]. An anti-actin antibody (product no. A5316, Sigma-Aldrich, St. Louis, MO, USA) 1:5000 at 5% BSA/TBST [*w*/*v*] was used as a loading control. WesternBright Quantum HRP substrate (product no. K-12043 D20, (LOT# 048M4843V); Advansta, Inc., Menlo Park, CA, USA) was used to detect chemiluminescence signals and the signals visualised using the Chemidoc XRS + System (Bio-Rad, Hercules, CA, USA). Quantification of visible bands was performed using a densitometer and ImageJ software (US National Institutes of Health, Bethesda, MD, USA).

### 4.6. Immunofluorescence

COCs and oocytes fixed in 4% paraformaldehyde [*v*/*v*] were washed in PBS then incubated for 15 min in 0.1 M glycine/PBS and permeabilised with 0.15% Triton X-100 [*v*/*v*] in PBS containing 1% BSA [*w*/*v*] for 15 min. Nonspecific binding sites were blocked by incubating in 2% BSA/PBS [*w*/*v*] for 15 min. Subsequently, oocytes and COCs were incubated for 60 min with anti-vaspin antibody 1:100 or anti-GRP78 antibody 1:50 (product no. PA5-30989 (LOT# TK26665551N), or PA5-19503 (LOT# GR32579381), respectively; Invitrogen, Carlsbad, CA, USA) at 1% BSA/PBS [*w*/*v*] for 1 h at room temperature. Next, oocytes were washed and incubated with anti-rabbit Alexa 488 antibody 1:500 at 1% BSA/PBS [*w*/*v*] (product no. A11008, (LOT# 1937184); Invitrogen, Carlsbad, CA, USA) for 1 h at room temperature protected from light. Negative control was performed by omitting the addition anti-vaspin and anti-GRP78 antibodies and by replacement of primary antibody by immunoglobulin G (IgG). After a further wash, observation was by fluorescent Axioplan Zeiss microscope.

### 4.7. DAPI and Hoechst 33342 Staining

For each experimental group, 25–30 oocytes were analysed for their nuclear status. After maturation, COCs were denuded by pipetting of COCs with 0.5% hyaluronidase [*v*/*v*] and the cumulus-free oocytes fixed in 4% paraformaldehyde [*v*/*v*] and then subjected to increasing (25–100%) concentrations of DAPI (product no. H-1200-10, Vector Laboratories, San Francisco, CA, USA) in 0.1% PVP/PBS [*v*/*v*] and the oocytes finally mounted on microscopic slides. To assess nuclear maturation status by Hoechst 33342, the fixed oocytes were maintained in PBS for 1 h and then mounted on microscope slides with 2.5 µg/mL of Hoechst 33342 (product no. 62249, ThermoFisher Scientific, Waltham, MA, USA). Observations were by Axioplan Zeiss fluorescence microscope. Results were expressed as the percentage of oocytes in metaphase-I and metaphase-II stages. The presence of the first polar body was taken as evidence of reaching the metaphase-II stage.

### 4.8. ELISA Assay

The concentration of P4 in the culture medium was determined by enzyme immunoassay using a commercially available ELISA kit (product no. EIA-1561, DRG Instruments GmbH, Marburg, Germany). The sensitivity of assay was 0.045 ng/mL, while the inter- and intra-experimental coefficients of variation were <9.96% and <6.99%, respectively. Each treatment was conducted in duplicate. Absorbance was measured using an ELx808 ELISA microplate reader and KC JUNIOR software at 405 nm wavelength (BioTek Instruments, Winooski, VT, USA).

### 4.9. Statistical Analysis

Results from three independent experiments were shown as mean ± SEM. For one experiment, COCs were collected from 100 pigs (15 follicles/ovary) and allocated after morphological examination to the experiments (30–50/group/experiment). The Shapiro-Wilk test was used to check the normality of the distribution. Statistical analysis was performed by t-test or one-way ANOVA, and the post-hoc Tukey test was used (PRISM software version 5; GraphPad, La Jolla, CA, USA). Significance was indicated by * *p* < 0.05, ** *p* < 0.01 and *** *p* < 0.001.

## 5. Conclusions

In conclusion, the data obtained first showed vaspin and GRP78 expression in porcine COCs, as well as the stimulatory effect of vaspin on porcine oocyte maturation in vitro by activation of MAP3/1 and PRKAA1 kinase pathways (Figure 6), confirming the regulatory role of vaspin in female fertility. The present study clearly links oocyte maturation with energy resources by direct the influence of the adipose tissue hormone, vaspin, on this process. The research also holds out hope of improving in vitro fertilisation protocols, which will help in solving the problem of infertility, especially for obese female. Moreover, the results obtained partly explain the link between the reproductive success of pigs and nutritional status or energy resources, which will give the possibility of manipulating pig breeding in the future in order to preserve endangered pig breeds and increase the economic profit of breeders. Knowledge about the impact of vaspin on in vitro oocyte maturation gives reason to suspect that this adipokine is involved in the development and differentiation of fertilised oocytes and early embryo development. This phenomenon has been observed previously; adiponectin stimulates not only porcine oocyte maturation but also the development of porcine embryos to the blastocyst stage [9]. In our future experiments, we plan to investigate vaspin expression in fertilised oocytes and to study its effect on genes participating in folliculogenesis, steroidogenesis, and early embryogenesis transcription, possibly by vaspin knockout. Moreover, new experiments may identify the mechanism of infertility connected with premature ovarian failure and the possible role of vaspin.

## Figures and Tables

**Figure 1 ijms-21-09342-f001:**
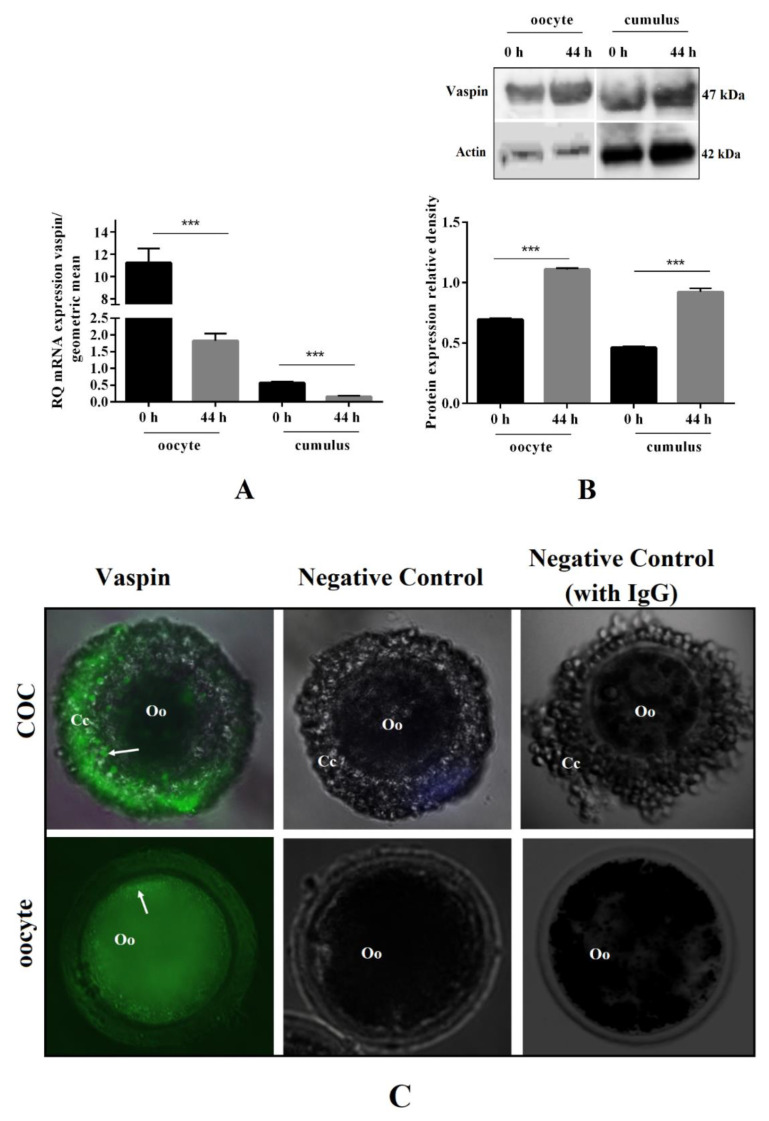
Vaspin mRNA and protein expression before and after oocyte in vitro maturation, as well as its immunolocalisation in cumulus–oocyte complexes (COCs). COCs (50/group/experiment) were selected after morphological examination of material collected from the ovaries taken from 100 pigs (15 follicles/ovary). COCs were collected before (0 h) and after (44 h) in vitro maturation and oocytes were denuded by gently pipetting in hyaluronidase, then real-time PCR and Western blot analysis were performed to determine mRNA (**A**) and protein (**B**) level of vaspin in oocytes and cumulus cells. Additionally, immunolocalisation of vaspin was analysed by immunofluorescence in COCs (**C**). Gene expression level was normalised to the geometric mean of actin and glyceraldehyde 3-phosphate dehydrogenase (GAPDH), and protein to actin. Vaspin immunostaining shown by arrows. Experiments were performed independently and repeated three times (*n = 3*). The data are plotted as the mean ± standard error of the mean (SEM) of three independent experiments. Significance between groups before and after maturation is indicated by *** *p* < 0.001; Cumulus cells (Cc), oocyte (Oo), Immunoglobulin G (IgG).

**Figure 2 ijms-21-09342-f002:**
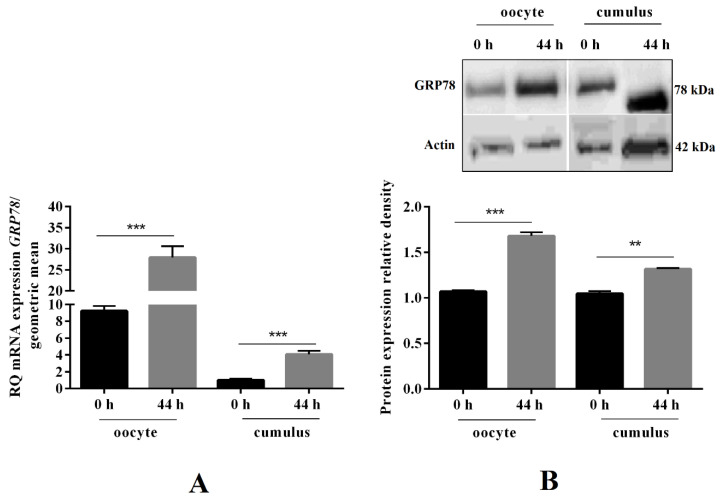
78-kDa glucose-regulated protein (GRP78) mRNA and protein expression before and after oocyte in vitro maturation, as well as its immunolocalisation in cumulus–oocyte complexes (COCs). COCs (50/group/experiment) were selected after morphological examination of material collected from the ovaries taken from 100 pigs (15 follicles/ovary). COCs were collected before (0 h) and after (44 h) in vitro maturation and oocytes were denuded by gently pipetting in hyaluronidase, then real-time PCR and Western blot analysis were performed to determine mRNA (**A**) and protein (**B**) level of GRP78 in oocytes and cumulus cells. Immunolocalisation of GRP78 was studied by immunofluorescence in COCs (**C**). Gene expression level was normalised to the geometric mean of actin and glyceraldehyde 3-phosphate dehydrogenase (GAPDH), and protein to actin. GRP78 immunostaining shown by arrows. Experiments were performed independently and repeated three times (*n* = 3). The data are plotted as the mean ± SEM of three independent experiments. Significance between groups before and after maturation is indicated by ** *p* < 0.01 and *** *p* < 0.001; Cumulus cells (Cc), oocyte (Oo), Immunoglobulin G (IgG).

**Figure 3 ijms-21-09342-f003:**
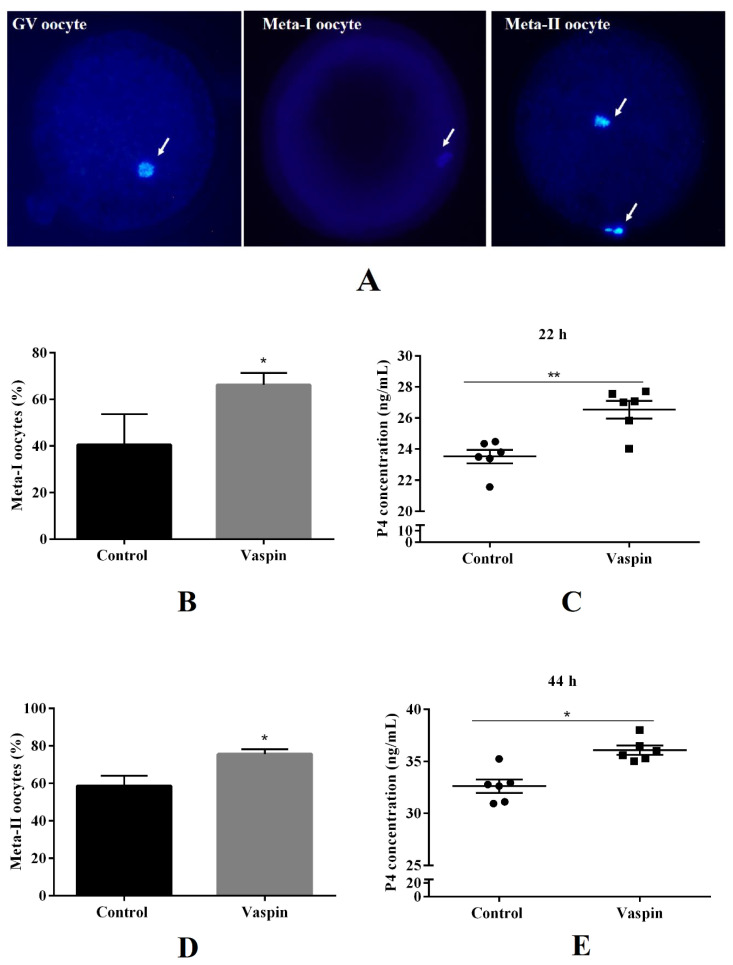
Effect of vaspin on nuclear oocyte maturation and progesterone (P4) secretion. COCs (30/group/experiment) were selected after morphological examination of material collected from the ovaries taken from 100 pigs (15 follicles/ovary). Porcine cumulus–oocyte complexes (COCs) were cultured for 22 h or 44 h in maturation medium in the presence or absence of vaspin (1 ng/mL) then mechanically separated into oocytes and cumulus cells. Nuclear maturation stages (**A**) were analysed by Hoechst 33342 or DAPI staining (**B**,**D**), while medium was collected and P4 levels measured by ELISA (**C**,**E**). DNA content shown by arrows. Experiments were performed independently and repeated three times (*n* = 3). Data are plotted as the mean ± SEM of three independent experiments. Significance between control and vaspin group is indicated by * *p* < 0.05 and ** *p <* 0.01; GV (germinal vehicle stage), metaphase-I of meiosis (meta-I), metaphase-II of meiosis (meta-II).

**Figure 4 ijms-21-09342-f004:**
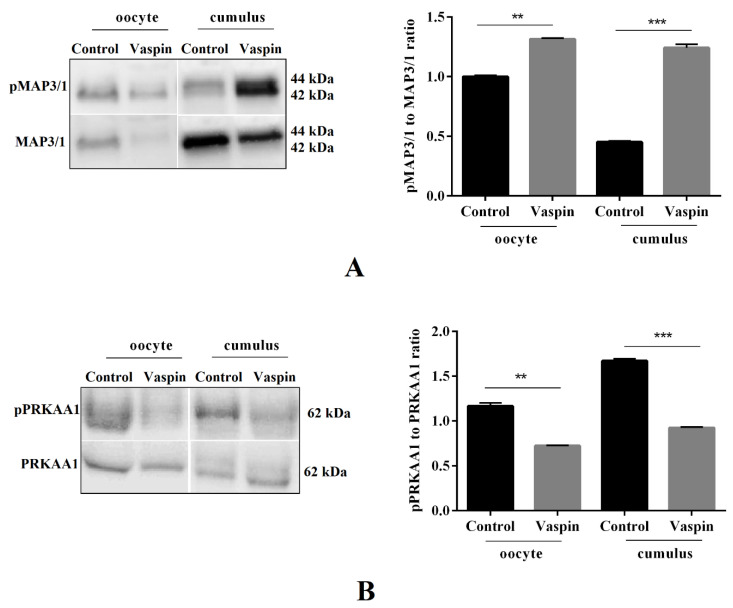
Effect of vaspin on mitogen-activated kinase (MAP3/1) and AMP-activated kinase (PRKAA1) phosphorylation in oocytes and cumulus cells after in vitro maturation. COCs (50/group/experiment) were selected after morphological examination of material collected from the ovaries taken from 100 pigs (15 follicles/ovary). Porcine cumulus–oocyte complex (COCs) were cultured for 44 h in maturation medium in the presence or absence of vaspin (1 ng/mL) then mechanically separated into oocytes and cumulus cells. Phospho MAP3/1 (pMAP3/1), MAP3/1 (**A**) and phospho PRKAA1 (pPRKAA1), PRKAA1 (**B**) protein expressions were analysed by Western blot analysis. Protein expression levels were normalised to actin. Experiments were performed independently and repeated three times (*n* = 3). Data are plotted as the mean ± SEM of three independent experiments. Significance between control and vaspin is indicated by ** *p* < 0.01 and *** *p* < 0.001.

**Figure 5 ijms-21-09342-f005:**
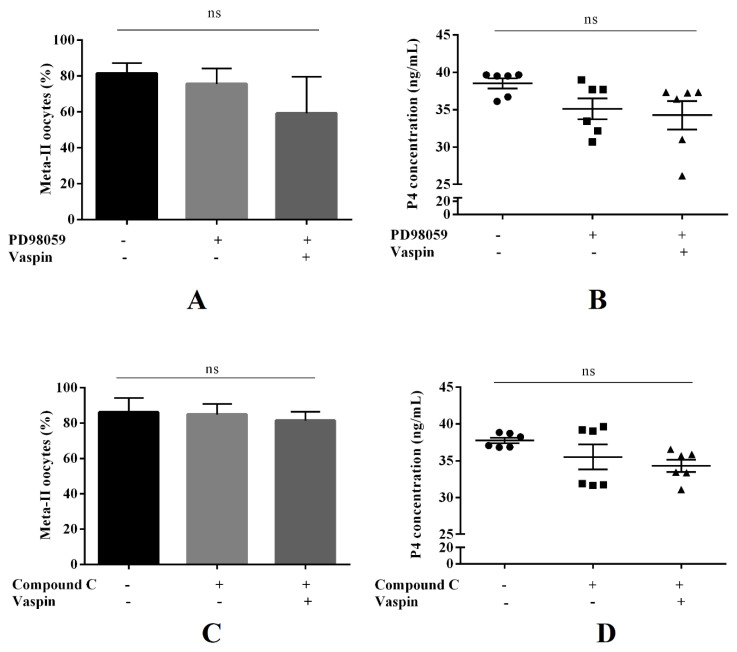
Involvement of the mitogen-activated kinase (MAP3/1) and AMP-activated kinase (PRKAA1) in the effect of vaspin on in vitro oocyte maturation. COCs (50/group/experiment) were selected after morphological examination of material collected from the ovaries taken from 100 pigs (15 follicles/ovary). Porcine cumulus–oocyte complexes (COCs) were cultured for 44 h in maturation medium with PD98059 (100 μM), MAP3/1 kinase inhibitor or Compound C (1 μM), PRKAA1 kinase inhibitor alone or with vaspin (1 ng/mL). The COCs were separated mechanically into oocyte and cumulus cells and nuclear maturation stages analysed by DAPI staining (**A**,**C**), while the medium was collected and progesterone (P4) level measured by ELISA (**B**,**D**). Experiments were performed independently and repeated three times (*n* = 3). Data are plotted as the mean ± SEM of three independent experiments. Statistical analysis was carried out at *p <* 0.05; metaphase-II of meiosis (meta-II), no significant (ns).

**Figure 6 ijms-21-09342-f006:**
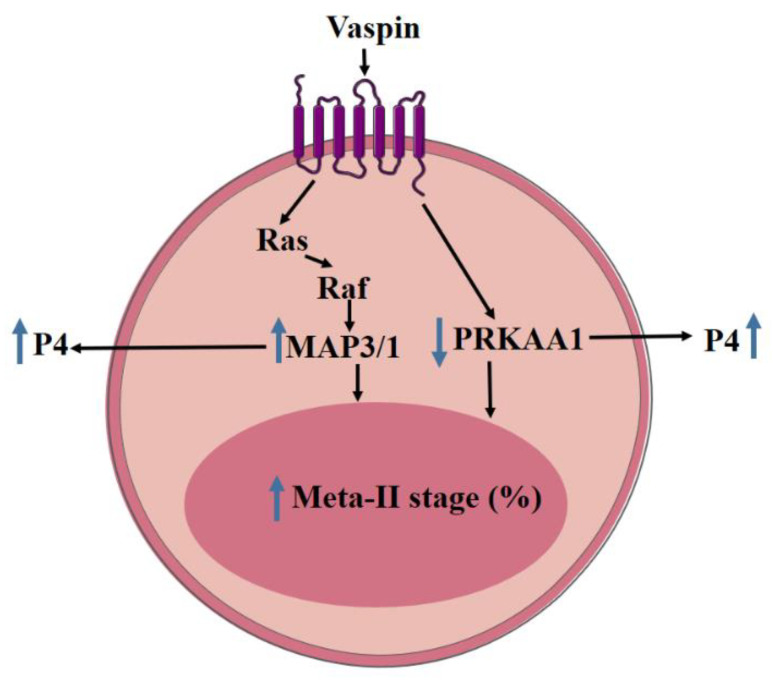
Model of the action of vaspin on in vitro porcine oocyte maturation. Vaspin stimulates oocyte nuclear maturation as well as progesterone (P4) secretion by activation of mitogen-activated kinase (MAP3/1) and inhibition of AMP-activated kinase (PRKAA1); metaphase-II of meiosis (meta-II); ↑ increase; ↓ decrease.

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
