# Peer review of "Expression and Impact of Vaspin on In Vitro Oocyte Maturation through MAP3/1 and PRKAA1 Signalling Pathways"

_ijms, 2020, doi:10.3390/ijms21249342_

Round 1
Reviewer 1 Report
Dear authors,
This manuscript is good and clear for the reader. However, I suggest you to add more images in both figure 1 and 2 as the comparison.
Figure 1. Image of immature oocyte stained with Vaspin and image of un-denuded mature oocyte stained with Vaspin
Figure 2. Image of immature oocyte stained with GRP78 and image of un-denuded mature oocyte stained with GRP78
Those additional images will give detailed information that either Vaspin and GRP78 moves from cumulus cell to cytoplasm.
Author Response
Reviewer 1:
We thank Reviewer for his/her thorough review and comments. Please find below our response to concerns regarding the present work.
This manuscript is good and clear for the reader. However, I suggest you to add more images in both figure 1 and 2 as the comparison.
Figure 1. Image of immature oocyte stained with Vaspin and image of un-denuded mature oocyte stained with Vaspin
Figure 2. Image of immature oocyte stained with GRP78 and image of un-denuded mature oocyte stained with GRP78
Those additional images will give detailed information that either Vaspin and GRP78 moves from cumulus cell to cytoplasm.
Our aim was to show the general vaspin/GRP78 immunolocalisation in COCs and denuded oocyte and then we focused on changes in it mRNA and protein in both oocytes and cumulus cells. We are agree that more images added to the manuscript will explain if vaspin/GRP78 moves from cumulus cells to cytoplasm during in vitro maturation. Unfortunately, due to COVID-19 pandemic it is impossible to prepare new experiments because of the restriction in slaughterhouses. However, according to our results, we can hypothesize that vaspin or GRP78 immunolocalisation doesn’t change during in vitro maturation because this same pattern of expression changes observed in oocytes and cumulus. We discuss this hypothesis in the discussion section.
We would like to thank Reviewer for time and effort reviewing this manuscript and positive comments. I believe that the valuable hints will raised the value of this scientific publication. I hope, that added explanations at least somewhat solved the problem raised.
Reviewer 2 Report
In their manuscript Kurowska and colleagues describe the expression of the adipokine vaspin and its receptor GRP78 in cumulus granulosa cells (CGCs) and denuded oocytes. Next authors investigate the role of vaspin on oocyte maturation in vitro and CGC progesterone production as well as the possible role of the MAP3/1 and PRKAA1 signalling pathways in these processes. The experiments are in general well performed, data presented are interesting and a logical continuation of previous work by the same group.
Comments:
1. The manuscript needs some serious English language editing by a native English-speaking scientist with knowledge of the research field. The text is at present in some cases really difficult to interpret due to language issues.
2. Although authors in general terms indicate in the Materials and Methods how many cumulus oocyte-complexes (COCs) were used per experiment, it is unclear whether these COC’s in one experiment were derived from one gilt or more gilts, and how COCs when coming from different gilts were divided over the different experimental groups. Are the average values shown in the figures the means of all oocytes, COCs or CGCs from the 3 experiments or are the averages of each experiment combined and thus are the means of these averages shown. Authors need to make this clear for each experiment preferably not only in the Materials and Methods but also in the figure legends. The way data are presented will influence the type of statistical analysis of the data.
3. In their Discussion authors speculate about how vaspin affects oocyte maturation, either directly or indirectly by affecting progesterone production (L.240-244). At present authors only show that there may be an association between progesterone, vaspin and oocyte maturation. An additional experiment in which COC’s are concomitantly treated with vaspin, an inhibitor of MAP3/1 and a progesterone receptor antagonist or CYP11A1/HSD3B inhibitor would make clear whether vaspin affects oocyte maturation directly or via inhibition of progesterone production. This experiment would be a valuable addition to the present study and make it more complete. As the number of data presented is not extensive in the present manuscript, I would urge the authors to perform this additional experiment.
4. Abstract
L.20 Immunolocalization by qPCR?
L.22 Please indicate the animal species from which the COCs were isolated and treated in vitro.
5. Results
L.184 Figure 5 p<0.05? Where does this refer to. This is not indicated in the figure or its legend. Neither does Figure 5 show a significant effect on cumulus expansion.
6. Discussion
In the first experiments authors investigate vaspin and GRP78 expression in CGCs and oocytes. This is followed by a possible role of vaspin in oocyte maturation and exposure of COCs etc to vaspin and the respective inhibitors. These experiments should be linked. What would be the physiological role of ovarian produced vaspin and how does this relate to peripheral vaspin levels and sow adiposity?
7. Materials and Methods
L.310-311 Why were ovaries from prepubertal gilts used and not from sows? Oocyte quality and redox balance in prepubertal gilts is compromised compared to sows.
L.374-376 and 392 Please provide the lot numbers of the antibodies used as well.
L396 This negative control is a control to test the specificity of the secondary antibody, not of the primary antibody. Please include the prober control, for instance in which the primary antibody has been replaced by isotype IgG.
8. Conclusion
The Conclusion could be written in a more focused way. Speculation of vaspin being a biomarker for PCOS or diabetes seems out of place here.
Author Response
Reviewer 2:
In their manuscript Kurowska and colleagues describe the expression of the adipokine vaspin and its receptor GRP78 in cumulus granulosa cells (CGCs) and denuded oocytes. Next authors investigate the role of vaspin on oocyte maturation in vitro and CGC progesterone production as well as the possible role of the MAP3/1 and PRKAA1 signaling pathways in these processes. The experiments are in general well performed, data presented are interesting and a logical continuation of previous work by the same group.
We thank Reviewer for his/her thorough review and comments. Please find below our response to concerns regarding the present work.
Comments:
- The manuscript needs some serious English language editing by a native English-speaking scientist with knowledge of the research field. The text is at present in some cases really difficult to interpret due to language issues.
As suggested by the reviewer, an English editing was performed.
- Although authors in general terms indicate in the Materials and Methods how many cumulus oocyte-complexes (COCs) were used per experiment, it is unclear whether these COC’s in one experiment were derived from one gilt or more gilts, and how COCs when coming from different gilts were divided over the different experimental groups. Are the average values shown in the figures the means of all oocytes, COCs or CGCs from the 3 experiments or are the averages of each experiment combined and thus are the means of these averages shown. Authors need to make this clear for each experiment preferably not only in the Materials and Methods but also in the figure legends. The way data are presented will influence the type of statistical analysis of the data.
We added in the manuscript all the missing information.
For the statistical analyses, on the graphs we showed mean of three culture results: mean of percent of oocytes in metaphase II (from 3 culture) or mean of P4 level (from 3 culture). For one experiment COCs were collected from 100 pigs (15 follicles/ovary) and after morphological examination COCs were selected to experiments (30-50/group/experiment).
- In their Discussion authors speculate about how vaspin affects oocyte maturation, either directly or indirectly by affecting progesterone production (L.240-244). At present authors only show that there may be an association between progesterone, vaspin and oocyte maturation. An additional experiment in which COC’s are concomitantly treated with vaspin, an inhibitor of MAP3/1 and a progesterone receptor antagonist or CYP11A1/HSD3B inhibitor would make clear whether vaspin affects oocyte maturation directly or via inhibition of progesterone production. This experiment would be a valuable addition to the present study and make it more complete. As the number of data presented is not extensive in the present manuscript, I would urge the authors to perform this additional experiment.
We showed that oocytes cultured with vaspin produced more P4, so this could explain a more efficient in vitro oocyte maturation, because during in vitro maturation P4 is produced by cumulus cells. On the other hand it is known that P4 alone stimulated oocyte in vitro maturation. We did not investigate whether vaspin stimulated in vitro oocyte maturation directly or indirectly by increasing P4 production and future studies are necessary to more precisely determine the molecular mechanisms involved. We suppose that vaspin action is direct because we observed its stimulatory action on MAP3/1 and inhibitory action on PRKAA1, which regulate oocytes maturation. Moreover, vaspin action was reversed by blocking these kinases, as well as reversed was P4 level. One kinase involved in P4 synthesis pathway is PKA, so decreasing in P4 level wasn’t connected with its inhibition, but with lower maturation rate. Moreover as other authors suggest only high doses of P4 like 500 ng/mL (Dode MA. Et al. Involvement of steroid hormones on in vitro maturation of pig oocytes. Theriogenology. 2002 Jan 15;57(2):811-21. doi: 10.1016/s0093-691x(01)00700-2.) stimulated porcine oocytes nuclear maturation, with no effect in smaller doses. In our experiment we observed, that P4 level increased by 3-4 ng/mL compared to control, so we postulated that is connected with it more efficient production by elevated percentage of mature oocytes, as well as this increase in P4 concentration is not efficient to stimulate porcine oocytes maturation alone.
On the other hand, it was observed in porcine oocytes that isoflavone daidzein inhibited production of P4 but does not affect oocytes maturation (Galeati G et al. 2010 Daidzein does affect progesterone secretion by pig cumulus cells but it does not impair oocytes IVM. Theriogenology. 2010 Aug;74(3):451-7. doi: 10.1016/j.theriogenology.2010.02.028), suggesting that vaspin could act similarly. New considerations were added to the discussion section.
New experiment will be for sure very interesting, but due to COVID-19 it is impossible to prepare new kind of experiment due to changing restriction in slaughterhouses. For this experiment we need minimum 3 experimental groups (control, P4 inhibitor, vaspin+ P4 inhibitor) so it is around 90 oocytes per single experiment. To obtain 90 oocytes, we have to collect ovaries from 100 pigs. Minimal repetition, what we have to prepare is 3, so 300 animals will be required. Obtaining this amount of material, its correct collection during slaughter and transport to the laboratory is limited by the current pandemic. We hope that expanding the discussion with new insights will at least somewhat resolve the problem.
- Abstract
L.20 Immunolocalization by qPCR?
We are sorry about this mistake.
We changed like this:
- i) the mRNA and protein expression of vaspin and its receptor 78-kDa glucose-regulated (GRP78) in porcine cumulus-oocyte complex (COCs), by real-time PCR and Western blot, respectively and their localization by immunofluorescence.
L.22 Please indicate the animal species from which the COCs were isolated and treated in vitro.
It has been added to the manuscript.
- Results
L.184 Figure 5 p<0.05? Where does this refer to. This is not indicated in the figure or its legend. Neither does Figure 5 show a significant effect on cumulus expansion.
We carried out statistical analyses at p<0.05 and we did not observe any significant difference, it was corrected in the manuscript. We also removed part about cumulus expansion, these results were just a microscopically observation.
- Discussion
In the first experiments authors investigate vaspin and GRP78 expression in CGCs and oocytes. This is followed by a possible role of vaspin in oocyte maturation and exposure of COCs etc to vaspin and the respective inhibitors. These experiments should be linked. What would be the physiological role of ovarian produced vaspin and how does this relate to peripheral vaspin levels and sow adiposity?
Discussion section was completed for new information.
- Materials and Methods
L.310-311 Why were ovaries from prepubertal gilts used and not from sows? Oocyte quality and redox balance in prepubertal gilts is compromised compared to sows.
The choice of our research model was based on previous papers describing effects on different hormones on porcine in vitro oocyte maturation, as follow:
- Zhang K. et al. Effects of ghrelin on in vitro development of porcine in vitro fertilized and parthenogenetic embryos. J Reprod Dev. 2007 Jun;53(3):647-53. doi: 10.1262/jrd.18140.
- Chappaz E. et al. Adiponectin enhances in vitro development of swine embryos. Domest Anim Endocrinol. 2008 Aug;35(2):198-207. doi: 10.1016/j.domaniend.2008.05.007.
And as indicated by Marchal: although prepubertal gilt oocytes appeared less meiotically and competent than their adult counterparts, they can be used to produce blastocysts able to develop to term. (Marchal R at al. Meiotic and developmental competence of prepubertal and adult swine oocytes. Theriogenology. 2001 Jul 1;56(1):17-29. doi: 10.1016/s0093-691x(01)00539-8.). This citation was added to the methodology section.
L.374-376 and 392 Please provide the lot numbers of the antibodies used as well.
The lot numbers were provided.
L396 This negative control is a control to test the specificity of the secondary antibody, not of the primary antibody. Please include the prober control, for instance in which the primary antibody has been replaced by isotype IgG.
The new control (with isotype IgG) were provided and in material and method section this information was added.
- Conclusion
The Conclusion could be written in a more focused way. Speculation of vaspin being a biomarker for PCOS or diabetes seems out of place here.
Information about PCOS and diabetes were removed. We added new information about adipokines impact on embryonic development and possible involvement of vaspin, as well as idea for future research.
We would like to thank Reviewer for time and effort reviewing this manuscript and positive comments. I believe that the valuable hints and reviews have raised the value of scientific publications and has become clearer.
Round 2
Reviewer 2 Report
Dear dr. Rak,
I am in general satisfied with your response to my concerns. It is a pity it is not possible to perform the additional experiment as it would have clearly contributed to the quality of the manuscript and the role of locally produced vaspin, or the combination of peripheral and ovarian produced vaspin, but I understand that the trouble you will have to obtain access to sufficient porcine ovaries. Your arguments do contribute to our understanding.
Could you please make clear in the Discussion of your manuscript your choice for gilts as well, as you did in your response to my comment.
Although the English of your text has improved, there is still moderate English language editing necessary.
Author Response
Thank you so much for your comments. We completed Discussion section as your suggestion.
As suggested by the reviewer, an English editing was performed before (R1) and certificate was added.